# Non-Invasive Fish Biometrics for Enhancing Precision and Understanding of Aquaculture Farming through Statistical Morphology Analysis and Machine Learning

**DOI:** 10.3390/ani14131850

**Published:** 2024-06-21

**Authors:** Fernando Joaquín Ramírez-Coronel, Oscar Mario Rodríguez-Elías, Edgard Esquer-Miranda, Madaín Pérez-Patricio, Anna Judith Pérez-Báez, Eduardo Antonio Hinojosa-Palafox

**Affiliations:** 1Division of Graduate Studies and Research, Tecnológico Nacional de México/ I.T. de Hermosillo, Av. Tecnológico 115, Hermosillo 83170, Sonora, Mexico; d09330874@hermosillo.tecnm.mx (F.J.R.-C.); eduardo.hinojosap@hermosillo.tecnm.mx (E.A.H.-P.); 2Licenciatura en Agro Negocios, Campus Hermosillo, Universidad Estatal de Sonora, Av. Ley Federal del Trabajo S/N, Hermosillo 83100, Sonora, Mexico; 3Division of Graduate Studies and Research, Tecnológico Nacional de México/ I.T. de Tuxtla Gutiérrez, Carretera Panamericana Km. 1080, Tuxtla Gutiérrez 29050, Chiapas, Mexico; madain.pp@tuxtla.tecnm.mx; 4Instituto de Acuacultura del Estado de Sonora, Conmonfort SN, Proyecto Rio Sonora Hermosillo XXI, Hermosillo 83280, Sonora, Mexico; anna.perez@sonora.gob.mx

**Keywords:** fish biometrics, fish biomass estimation, fish morphology, computer vision, image processing, machine learning

## Abstract

**Simple Summary:**

This research addresses the problem of biomass estimation and fish development monitoring by developing and meticulously validating a computer-vision-interpretable methodology for fish biometrics based on the extraction of statistical features of shape and size through a signature function. This is done by comparing manually extracted features in machine learning predictions against the computer vision (image feature extraction) machine learning predictions. This research shows promising results in statistical accuracy and error metrics, especially the computer vision methodology plus artificial neural networks for biomass prediction. It is concluded that the signature-function-based methodology plus the neural networks regression is competitive for biomass estimation and provides a powerful interpretable tool for species morphology development research. As a result, from a better understanding of fish development and biomass estimation, the aquaculture sector can improve production, wasting less animal food and detecting fish welfare problems through deformity detection, which translates into reduced costs and improved quality of animal meat.

**Abstract:**

Aquaculture requires precise non-invasive methods for biomass estimation. This research validates a novel computer vision methodology that uses a signature function-based feature extraction algorithm combining statistical morphological analysis of the size and shape of fish and machine learning to improve the accuracy of biomass estimation in fishponds and is specifically applied to tilapia (*Oreochromis niloticus*). These features that are automatically extracted from images are put to the test against previously manually extracted features by comparing the results when applied to three common machine learning methods under two different lighting conditions. The dataset for this analysis encompasses 129 tilapia samples. The results give promising outcomes since the multilayer perceptron model shows robust performance, consistently demonstrating superior accuracy across different features and lighting conditions. The interpretable nature of the model, rooted in the statistical features of the signature function, could provide insights into the morphological and allometric changes at different developmental stages. A comparative analysis against existing literature underscores the competitiveness of the proposed methodology, pointing to advancements in precision, interpretability, and species versatility. This research contributes significantly to the field, accelerating the quest for non-invasive fish biometrics that can be generalized across various aquaculture species in different stages of development. In combination with detection, tracking, and posture recognition, deep learning methodologies such as the one provided in the latest studies could generate a powerful method for real-time fish morphology development, biomass estimation, and welfare monitoring, which are crucial for the effective management of fish farms.

## 1. Introduction

The farming of aquatic species, known as aquaculture, plays a pivotal role in meeting the increasing global demand for seafood. In 2020, aquaculture contributed to 49.2% of the total production of aquatic animals, and its significance is expected to grow in the coming years [1]. Amongst the challenges posed by climate change, a growing world population, and the depletion of fish stocks, enhancing the efficiency and sustainability of aquaculture is necessary. Central to this endeavor is the crucial need to address the significant issue of feeding, which could constitute up to approximately 80% of the total operational costs of aquaculture [2,3]. Accurate fish biometrics, particularly non-invasive methods for biomass estimation, arise as a crucial component in optimizing feeding strategies, thereby mitigating overfitting and underfitting feeding issues that can lead to environmental contamination, potential fish health problems, and production losses.

Within the realm of aquaculture, accurate biomass estimation is capital for effective pond management. Traditional methods involve invasive sampling, wherein fish are taken out of the water, measured, and weighed. However, these approaches are labor-intensive, prone to injury, and cause stress to the fish [4]. Stress in aquaculture is a multifaceted problem that can compromise fish well-being, growth, reproductive capacity, and overall survivability [5,6,7]. As such, there is a clear need for non-invasive technologies to advance the precision of biomass estimation while minimizing the impact on fish welfare. Most of the studies related to fish biometrics in aquaculture estimate size or count fish in images, but only a few tackle the problem of biomass or weight estimation. There are studies on precision livestock farming of biomass estimation from images of cattle and pigs [8,9,10], but little has been done in the analogous area of precision aquaculture, which, as stated by [11], seeks to apply control engineering concepts to the cultivation of aquatic species and one of its four fundamental objectives is to monitor animal biomass in a more autonomous way.

Existing literature highlights the limitations of traditional methods in estimating total pond biomass, mainly due to their invasive nature. These drawbacks include the potential for stress-induced health issues in fish, chemical composition alterations in the rearing environment, and constraints in scalability [4,12,13]. The pressing research gap lies in developing non-invasive, scalable, and accurate methodologies for fish biometrics, providing a holistic understanding of morphological changes without compromising animal welfare and with interpretability and species generalizability capacities.

There is also a research gap in standardization and validation of the image processing techniques used for extracting morphometric parameters, as mentioned by [14]. Current studies often employ diverse image capture devices and methodologies, leading to variations in data quality and consistency [11,15]. Addressing this gap is crucial for establishing a robust and more universally applicable framework that ensures the reliability and comparability of results across different aquaculture settings and fish species since the current state of computer vision in aquaculture encompasses diverse methodologies for capturing devices and illumination, preprocessing, segmentation, and machine learning steps for different environment and species [11,15,16,17,18]. Developing standardized protocols for image acquisition, pre-processing, and feature extraction would significantly enhance the reproducibility and scalability of non-invasive fish biometric studies, fostering the broader adoption of this technology in the aquaculture industry [11].

This study addresses the research gap by assessing a novel approach for fish biomass estimation, focusing specifically on the validation of Nile tilapia (*Oreochromis niloticus*) since it is a common and widely cultivated farming species. The aim is to employ statistical morphological analysis and machine learning to develop a non-invasive, interpretative model for precise biomass estimation. By utilizing a signature function-based feature extraction methodology, which considers both size and shape traits, this research aims to contribute to the advancement of non-invasive biometrics in aquaculture. The significance of this study lies in its potential, in conjunction with new methods for real-time segmentation and preprocessing, to provide a scalable and welfare-conscious solution for optimizing feeding strategies and understanding species morphological development and its relationship with biomass growth. This, in turn, enhances the sustainability and efficiency of tilapia farming and aquaculture farming in general, thereby contributing to the broader field of fish biology.

The remainder of this paper is organized as follows: In Section 2, related works are described considering methodology improvements, and particularities, dataset captured conditions, and metrics. Section 3 describes the materials and methods used for the computer vision methodology based on the chord length function, focusing on feature extraction and machine learning models. Section 4 presents the results obtained using the machine learning models applied to the features extracted by the chord length function algorithm for biomass estimation. Section 5 discusses the presented results against the state-of-the-art models, and Section 6 gives the corresponding conclusions.

## 2. Related Works on Fish Biomass Estimation

Traditional methods for biomass estimation often rely on taking fish out of the water with fishing nets and measuring them with a balance and ruler, which stresses the animals and can cause diseases and death [16,19]. Moreover, having poor monitoring conditions in aquaculture farms can cause production losses and adverse environmental effects [20]. New advancements in image processing, computer vision, and artificial intelligence in general, are moving us closer to a non-invasive reliable method for fishpond biomass estimation in industrial setups, which can also be of help to biologists and aquaculturists in studying fish-grown development and morphological changes at distinct stages of life and are scalable across different species, which has not been reached and is currently a research gap and problem [11,18].

There is a huge diversity of setups that can be found in practice, and so too are the research studies that have tried to tackle the problem and the variations in species, conditions, camera location, and validation methods for computer vision development [16,17,18]. All these works use a methodology based on machine learning or deep learning whose basic steps are shown in Figure 1. Most works use varied methods for preprocessing, segmentation, and feature extraction; however, for biomass estimation, they use common linear, power, or polynomial models adjusted to the extracted data for biomass estimation; that is to say, they use computer vision to estimate simple features like length or weight, and then use ML or statistical models to associate those features with biomass [17].

Improvement of the methodologies can be done by improving one or more of the ML methodology’s steps or improving the deep learning architecture. Deep learning can also be used for preprocessing, segmentation, feature extraction, and fitting curves to correlate features with weight or for complete biomass estimation [16].

The subsequent related works are described in chronological order, highlighting the varied conditions that were used and the important aspects required for developing a non-invasive biomass estimation method, which also helps in studying fish morphological changes during aquaculture processes.

In [21], a method based on stereo vision (two cameras) is used to segment gilthead seabream in 3D using distinctive fish characteristics (frontal, rear, and intermediate points). To estimate the final weight, common length–weight relationships, well-known in aquaculture, are utilized. Specifically, the author employs a power model and validates the experiment with 122 specimens in breeding tanks, achieving errors of less than 4% for the system with cameras above water and less than 5% with underwater cameras. The author highlights the difficulty posed by reflections on the water surface when capturing images, prompting improvements in fish preprocessing and segmentation. While this model enables length extraction and yields good results in controlled environments, a single dimension is insufficient to capture the morphological changes occurring during fish growth stages.

In [22], the relationship between seven visual features (Length, Height, Area, Perimeter, Equivalent Diameter, and Major and Minor Axis Lengths) extracted from images of *Oncorhynchus mykiss* and weight applying linear and multiple regressions are evaluated. A 12 MP camera and a dataset of 75 specimens in a controlled light setup were used. Linear, logarithmic, exponential, and power models were also fit, and it was found that the power model based on area was the best fit, based on the R2 metric and standard error of estimate (SEE). Weight categories and the error percentage are also stated, from which insights can be found. Thus, this work is an ML methodology that tries to improve weight estimation by enhancing feature extraction and the ML step.

In [4], a novel monocular system in an underwater light-controlled setup with image enhancement filters (Homomorphic, CLAHE, Guided), a combination of 2D saliency and morphological operators for segmentation, and 3rd-degree polynomial regression on *Oreochromis niloticus* length are proposed. These authors used 150 fish specimens of *Oreochromis niloticus* and tested several regressor methods, finding that the grade-3 polynomial is the best, based on the following metrics: RMSE, MAE, MAPE, MSLE, EVS, and R2. The work proposes a single subaquatic camera in a controlled environment and achieves MAPE of 11%.

In [23,24], a model based on CNNs to segment the area corresponding to the fish was applied. To estimate biomass, they adjusted one and two-factor models using pairs obtained from area–weight. In (Konovalov, 2019) the previous work is continued by testing two-factor models and experimenting with cases of images without fins and images with fins. In this case, a Direct Weight from Area regressor CNN, LinkNet-34R, is trained and applied to a different test set from the two used for model development and images extracted with no fins. In the article, the average weight is not shown, only the ranges. In the conclusions, it is stated that the LinkNet-34R gave MAPEs of 4–11%, with 4.28% for images without fins and 11.4% for images with fins.

In [25], a low-cost stereo vision monitoring system with the Hugh gradient method and 3rd-degree polynomial-based weight estimation capabilities for Nile tilapia (*Oreochromis niloticus*) using Raspberry Pi and low-cost USB cameras is proposed. It extracts the contour and then follows a pixel/metric conversion to estimate length. Polynomial, exponential, linear, and logarithmic curves are adjusted to the data, with the 3-degree polynomial identified as the best option in terms of Mean Percentage Error (MPE), reaching a value of 2.82%. The fish are housed in a fishbowl and the camera is positioned outside, observing them from the side. Two setups are utilized, one for manual measurements and the other for automatic weighting with the stereo camera system. This is important to have a way of comparing the models with the ground truth.

In [26], linear regression models based on the area in pixels of fish-sea bass (*Lates calcarifer*) images taken from a high angle to estimate weights are applied. Twenty-five fish were placed in boxes with a water level of 7 cm and captured by an Olympus EM 10 Mark II camera (Tokyo, Japan) with a resolution of 4608 × 2692 pixels. Fish images were collected weekly for a month, with a train–test split of 40/60%. Ten images were taken of each fish, resulting in a total of 150 images used for testing each week and a total of 600 images for the full month. This research closely aligns with the present study in terms of methodology. The authors note that the high-angle camera position is chosen because the shape of fish fins changes due to various factors such as natural erosion or handling during photo capture. Metrics computed by the authors include root mean square error (RMSE), mean absolute error (MAE), mean absolute relative error (MARE), maximum absolute error (MXAE), and maximum relative error (MXRE). Among the results, an average MAE of 6.06 g, an average R2 of 0.77, and an average MARE of 5.18% were reported.

In [27], a power model to estimate weight from length is derived and used to develop a program for weight estimation from images. The program captures pictures in the aquaculture setup, and the user marks the fish by drawing a line from tip to tail. Subsequently, the program provides statistical measurements. In total, 270 fish were sampled every two weeks to obtain total length and weight. Pictures were taken with a focal length of 14 mm, exposure of 1/40 s, and a resolution of 4608 × 2592 pixels. The program requires an object of known size for calibration to establish the pixel-to-centimeter relationship. Additionally, the user is required to draw a line identifying the length of a fish. The program was used to examine 17 images taken by the developed program, resulting in an average accuracy of 93.01%, implying an average error of 6.99% for weight estimation.

In [28], a non-invasive 2D tilapia biomass estimation method utilizing a single low-cost camera to observe the fish is developed. The method consists of a tilapia detection step and a tilapia weight estimation step. Firstly, a Mask Recurrent Convolutional Neural Network model is trained to extract fish dimensions in pixels. Secondly, the depth of the fish is estimated, and thirdly, the dimensions of the extracted image are converted from pixels to centimeters. After these steps, weight is estimated using regression learning, testing different machine learning models to determine the best option for biomass estimation. It achieved mean absolute errors (MAEs) of 42.5 g and R2 of 0.70, with an average weight error of 30.3 in a turbid environment. The final methodology, named Tilapia Weight Estimation Deep Regression Learning (TWE-DRL), requires only three features: age, length, and width of the fish. This may be a limitation as it necessitates prior information on the age of the fish. The tilapia dataset was obtained from images and videos of tilapias in three tanks, each containing 30 tilapias. A total of 5037 images were taken at a resolution of 1920 × 1080 pixels. The fish are detected with their bodies horizontally aligned to the image.

In [29], Convolutional Neural Networks (CNNs) and transfer learning to estimate Nile tilapia biomass are used. Eight pre-trained ConvNet models were evaluated and modified, trained, validated, and tested with the given datasets. The photos were captured in a conditioned container with a light source, with a camera resolution of 2590 pixels × 1942 pixels. The power area model for the same dataset achieved a mean absolute error (MAE) of 2.69 g, while the modified VGG-19 model yielded an outstanding MAE of 2.27 g. However, the method involves taking the fish out of the water, which although for a brief period (CPU and GPU < 1 s), remains invasive and stressful for the animals. The evaluation of different modified models in this research involved metrics such as root mean square error (RMSE), MAE, mean relative error (MRE), mean absolute percentage error (MAPE), and R2.

In [30], a new method for shrimp biomass estimation using morphometric features based on underwater image analysis and a machine learning approach is developed. The single camera is calibrated using triangle similarity (TS) and correction factor (CF), resulting in a hybrid method based on the triangle of similarity, correction factor, and multilinear regression (TS-CF-MLR). The camera used has a resolution of 48 megapixels. The method was validated with a small number of samples, approximately twenty for training and six shrimp images for testing. Two lamps, one underwater and the other above the water, aided with shrimp detection. Calibration requires manual measurements of the shrimp. The method estimates results from images taken in a real pool with shrimps.

In [31], a buoy robotic system for estimating gilthead seabream length and weight using a stereoscopic camera, artificial intelligent algorithms, and computer vision is described. The images were captured at the end of the on-growing period. A power model was developed using 190 samples of seabream. The employed camera is an 8MP Arducam synchronized stereo camera with two 8MP IMX219 sensors. Evaluation metrics include RMSE, MAE, and R2, with the best model achieving outstanding results of 0.05, 0.04, and 0.96, respectively.

In [32], a novel methodology for real-time fish weight estimation, combining fish posture recognition using deep learning with biomass estimation using stereo vision and length–height–weight relationships is introduced. The metric used in this study was mean relative error (MRE), which reached 2.87%. The camera resolution was configured to be 1280 × 720 pixels, with a frame rate of 30 fps and a pixel size of 2 × 2 μm. The dataset for biomass estimation comprised measurements of weight, height, and total length obtained from 550 fish weighing between 150 and 750 g. The evaluation showed RMSE and MRE values between calculated and measured fish body weights of 35.97 g and 8.86% in the length–weight relationship (L-W) and 34.90 g and 7.41% in the height–weight relationship (H-W).

All these previous works contribute to various stages, as depicted in Figure 1, and demonstrate the diversity of conditions, species, number, and type of samples, as well as equipment used to achieve the obtained results. These methodologies have been applied by the authors to only one species, and most of them extract only a few features that could aid in the morphological development analysis of the life stages of a species. Others, such as [30], extract morphometric features that are very specific to one species, and [22] extract features that can be used in multiple species but do not contemplate many form parameters directly related to shape and not size only. Essentially, most of the works deal with allometric relationships of a fish under specific conditions and positions. Allometry is the relationship between size (length, height, area, etc.) and weight [33]. However, as evidenced in [32], different life stages of a species may require more features for better biomass estimation and for biological and aquaculture studies. Thus, a feature extraction method such as the one in [34], which provides not only size but also shape parameters of any closed contour, could be of help in the quest for better features and standardizing processes for non-invasive biometrics.

In [26], it is observed that capturing pictures from the top is a good choice because the shape of the fins changes due to several factors. Common experience suggests that a simple, interpretable, and scalable computer vision methodology is sought, especially for the most cultivated species like finfish, shrimps, and crabs. Recognizing them from a top-view is easier for a human being, considering the possibility of changing position and orientation, as most of them usually adopt a common and easily recognizable top-viewed position. An example of this is depicted in Figure 2, where in Figure 2A, several freshwater shrimps are seen from a camera that is at a low angle with respect to the bottom; here can be seen the different shapes that a shrimp image could take, depending on its orientation. It could be even a little difficult to identify a species with the naked eye simply by looking at the image of the animal from certain specific orientations. In Figure 2B, shrimps are seen from a high angle and a common shape can be observed, no matter the orientation of the shrimp with respect to the axis perpendicular to the bottom. By nature, fish, shrimp, and crabs which are also the most cultivated species, usually maintain a certain angle with respect to the plane parallel to the bottom and only abruptly change it when need to go deeper or go to the surface of the water. However, the angle with respect to the axis perpendicular to the bottom is very variable. Thus, a camera located at a high angle can be a good choice for developing a method that can work with different species, as shown in Figure 3, where characteristic shapes of fish, shrimp, and crabs can be easily detected by the naked eye. This could imply that a method would need to learn smaller quantities of images of shapes of species from a high angle than from a low angle.

## 3. Materials and Methods

This section begins by describing the tools (hardware and software) and methodologies that were used to sample the weights, lengths, heights, and images of tilapia. Section 3.1 describes the data collection hardware and software used for taking the pictures and measuring the lengths, heights, and weights of the tilapias. Section 3.2 describes the applied feature extraction algorithm developed by [34] and the threshold segmentation techniques used for fitting it to the contours of tilapias.

### 3.1. Data Collection

The data were collected from two different locations. A total of 112 samples were obtained from square pools of no more than 3 × 2 m and 17 samples were obtained from a circular pond (diameter = 10 m, depth = 1.5 m) and an Aquarium. Data for each sample consisted of the weight, length, height, and image of the fish.

The dataset used in this study to assess the computer vision methodology, specifically the feature extraction method stage and the machine learning stage for biomass estimation, comprises manually extracted features and images of Nile tilapia (*Oreochromis niloticus*) under varying conditions. The manually extracted features were the length, height, and weight of each tilapia, which were obtained by manually measuring the 129 fish with a ruler, vernier caliper, and balance, as illustrated in Figure 4. These features (length and height) were chosen since weight can be easily estimated from them and compared against the measured ground truth [19]. The specimens were then placed in a dry ice cooler with an installed Logitech 920c webcam (Lausanne, Switzerland) with a resolution of 1 MP, as shown in Figure 5. A program developed in Python language with the OpenCV, glob, and PIL libraries was written to automate the process of taking pictures. The program allowed us to take several pictures at time intervals given by the user. The parameters were set up by trial and error to ensure that good images were obtained.

A total of 129 samples (data and images), were collected, ensuring a diverse representation of tilapia specimens in aquaculture setups. A total of 17 samples were captured under varying lighting conditions while the other 112 samples were captured in a controlled light environment. The images were captured under non-invasive conditions, eliminating the need for physically handling the fish. The non-invasiveness of the data collection method is crucial for minimizing stress on the aquatic species and replicating real-world scenarios in aquaculture since the fish were in the water, although almost on the surface. The water level was 10 cm for the 112 samples and 15 cm for the 17 samples, with high light reflection.

### 3.2. Image Processing and Feature Extraction

The image processing methodology adopted in this study is founded on the chord length function whose concept was proposed by [35], providing a robust approach for capturing both size and shape traits of the tilapia specimens. Size and shape, which comprise what is called form, are the two main components of an object’s contour [36]. There are multiple parameters that try to quantify shape in an intuitive manner; these are called shape descriptors [37]. The chord length function is one of them. The chord length function enables the extraction of morphological features essential for biomass estimation. Statistical features, including mean, perimeter (sum of frequencies of the histogram of the contour), area (perimeter × mean) standard deviation, and mode of the inner perpendicular segments of the contours, are extracted from the processed images. These features serve as key input parameters for subsequent machine learning models.

The algorithm used for shape analysis is proposed by [34] and is described as Algorithm 1 (code was added to compute extra shape and size parameters):
**Algorithm 1**: Shape analysis using a chord length function**Input:** The *images*, *threshold* for every image, *n* (number of utilized neighbors)**Output:** container of perpendicular segment longitudes (histogram) and statistical parameters (mean, mode, median, maximum, minimum, standard deviation, perimeter, area).1.for image in the image_dataset:  convert to grayscale2.  **# Apply preprocessing and segmentation:**
  Threshold the image using the manually selected standard *threshold* parameter, opening, Canny’s Edge Detector, and Octal thinning  algorithm.3.  **# Exhaustive search for contours:**  create list_of_contours    for row in rows:    for column in columns: 4.      **#Store the pixels of the found contour in a list**      if pixel (row, column) is contour:        create a list to store the fish contour’s pixels        store the pixel’s coordinates in a list         while pixel_found = True:          search the 8-neighbors          if there is a neighbor and the neighbor is           different to the first pixel found:            go to the next neighbor          else:             pixel_found = False       store the contour (list) in list_of_contours5.  **# transform the bidimensional (row, column) pixel**  **representation to a unidimensional representation given by the**
  **perpendicular segment to the tangent line to the pixel for**
  **each contour**
  for each contour in list_of_contours:    create an empty histogram    for each pixel in the contour_list:      compute the n-connected neighbor       compute the secant line between the pixel and the n      connected neighbor      compute the perpendicular segment to the secant line       that is contained in the contour      store the perpendicular segment in histogram    store each histogram in list_of_histograms6.  **# plot the histogram and compute the statistical parameters**
  **for each tilapia image**  for each histogram in list_of_histograms    plot histogram    compute mean, median, mode, total_frecuency (perimeter),     Area (perimeter*mean), standard_deviation, maximum,     minimum, perimeter/mean

The algorithm begins converting the image to gray scale since color is not a relevant parameter for identifying the size of the tilapia and doing this allows us to simplify the problem. Afterward, a morphological opening process is applied to eliminate small objects floating in the water or small stains produced by some other cause. Then, a Canny edge detector, and finally, an octal thinner are applied to ensure that the edges are exactly one pixel wide. This is necessary in order to apply the feature extraction method based on the chord length function. After extracting the image with edges one pixel wide, a search for the contours in the image is carried out. When a contour is found, it is stored in a list, pixel by pixel. To do this, contour pixels are searched in the 8-neighbors of the current pixel until the contour is finished, either because the pixels have run out (open contour) or the initial pixel has been reached (closed contour), once the contour has been traversed it is stored in a list of contours. This is repeated each time a new contour is found until the entire image is covered. In the next step, the contour list is taken, and each contour is transformed into a list of interior perpendicular segments of the contour; that is, each pixel represented by two dimensions (row, column) is replaced by a single value that results from calculating the length of the line segment perpendicular to said pixel and that is contained in the contour. The list of lengths is stored. In turn, a histogram that represents each list of perpendicular segments, which represent the shape and size of the objects found in the image, is generated.

To the best of the knowledge of these authors, no work has validated a signature function-based size and shape feature extraction method in a computer vision pipeline for biomass estimation of aquaculture species.

To segment the fish from the environment to apply the chord length function feature extraction methodology, steps 3–6 of the algorithm are presented. A binarization methodology was chosen by visual inspection of the results of seven filters provided by the Python library sickit-image, with the function <<*try_all_threshold ()*>>. In Figure 6a, an example of a fish from the 112 fish in the light-controlled condition dataset is presented, and in Figure 6b, the same is shown with an example of a fish from the 17 fish in the non-light-controlled conditions dataset.

As can be seen in Figure 6b, the segmentation for some images from the 17 fish in the non-light-controlled environment was not good. This affected the precision of the full computer vision biomass estimation pipeline applied to the 129 fish images. Future research will be required to improve this task.

Figure 7 shows an example of how the feature extraction method works by applying it to a tilapia. This method provides a histogram representation of the form of the object given by its contour, from which statistical parameters for shape and size can be obtained. Figure 7a shows the grayscale image, Figure 7b shows the fish after segmentation, Figure 7c shows the contour, which is stored in a list or container to continue to the next step, Figure 7d,e show the progression of computing the perpendicular inner segments of the fish, Figure 7f shows a representation of the list of lengths of the perpendicular inner segments, and Figure 7g shows the resulting histogram for the form of the fish.

### 3.3. Machine Learning Models

Three distinct machine learning models chosen based on their complexity and used in the literature and practical applications were employed to predict tilapia biomass based on the extracted features:

Linear regression: This model captures the linear relationship between the morphological features and tilapia biomass. Its simplicity allows for interpretability, providing insights into the direct impact of individual features on biomass estimation. This model was used by [22,28] and compared against other common and novel alternatives. A linear model correlating area and weight was used in [26]; in terms of image acquisition, this is the setup closest to the present work.

Power model: The power model was selected to account for potential non-linear relationships between morphological features and biomass. This model introduces flexibility, accommodating complex patterns in the data that may not be captured by linear regression. Several works have shown the power model relationships between features like length and weight [22,27,31].

Multilayer perceptron (MLP): The MLP model, a neural network architecture, was chosen for its ability to learn intricate patterns and relationships between the data. The deep learning capabilities of the MLP make it suitable for handling complex, non-linear associations between image-derived features and tilapia biomass.

The reason for choosing these three models is that they provide low, medium, and high complexity models that allow us to get insights into the threats and drawbacks of the complexity–interpretability relationship and allow for model improvement. The models were applied to the manually and automated extracted features using the sklearn library in Python.

The experiments are divided into two steps:

Step 1:

In the first step, all three models were applied to the dataset with all 129 images (images taken under controlled and non-controlled light conditions) for each of the following inputs to estimate the weight (in grams, g):Length (in centimeters, cm) manually obtained using a ruler and vernier.Height (in centimeters, cm) manually obtained using a ruler and vernier.Length and height (in centimeters, cm) manually obtained using a ruler and vernier.Perimeter (in pixels) obtained using the image processing methodology.Area (in squared pixels) obtained using the image processing methodology.Mean (in pixels) obtained using the image processing methodology.Standard deviation (in pixels) obtained using the image processing methodology.Mode (in pixels) obtained using the image processing methodology.Perimeter, mean, standard deviation, and mode obtained using the image processing methodology.

The same was conducted using the dataset of 112 images taken under controlled-light conditions and compared to see how much illumination and variation in size affected the results.

The weight distribution histograms for the datasets are shown in Figure 8.

Both datasets exhibit positively skewed weight distributions, indicating a concentration of smaller weights with a tail of larger weights.

As the dataset is small, the machine learning models were applied 50,000 times for the linear regression and power model and 300 times for the multilayer perceptron to get the average MAE and R2. That is to say, the dataset was divided randomly into training and test sets (25% test set and 75% training set) in each iteration, and then the model was applied to that division of training and test sets. For each iteration, the mean absolute error (MAE) and the coefficient of determination (R2) were obtained. Finally, the average MEA and R2 were computed.

Step 2:

In the second step, the machine learning models were trained on the images in the whole dataset, without partitioning it into training and test sets. Thus, no iterations were performed. This was done with the goal of observing how well the entire dataset fit the commonly used curves in biomass estimation and not for generalization since that was done in step 1. The models were applied to the entire (not divided) dataset and the mean absolute error (MAE) and coefficient of determination (R2) were obtained for the model applied to the entire dataset for each feature (manually extracted length and height and automatically extracted (from the images) perimeter, area, mean, standard deviation, and mode).

The rationale behind employing diverse models lies in comprehensively exploring the trade-off between model complexity and interpretability, ultimately striving for an optimal solution for non-invasive biomass estimation.

The biomass estimation capability of the computer-vision-proposed methodology is validated by comparing it against a “manually extracted features machine learning methodology” using the most commonly used features (length and height) and the three most common regression algorithms in computer vision that are applied to fish (linear regression, power model, and multilayer perceptron). A diagram of the proposed methodology and how to validate it is shown in Figure 9.

## 4. Results

The results of the experiments validate the computer vision methodology and, more specifically, the statistical shape and size feature extraction capabilities in combination with the machine learning models. First, in Section 4.1, the automatic feature extraction biomass estimation methodologies are compared against the manually extracted features plus machine learning methodologies in the dataset containing all the samples. Then, the same is done for the 112 images taken under the controlled-light condition environment. An overview of the general results is provided, and finally, the machine learning models are fitted with the complete dataset as the training set to see how well the model fits all the data. In Section 4.2, the results of sensitivity against light conditions are described. In Section 4.3, the generalizability and consistency of the MLP model results across features and conditions are provided.

### 4.1. Comparative Performance of the Models across Features

This section presents the results of the comparison of models for different characteristics extracted both manually and automatically.

#### 4.1.1. Experiment Using Partitioned Dataset Containing All the Samples

In this case, the linear regression, power function, and multilayer perceptron models are applied to the manually and automatically extracted features in the whole dataset (129 fish). As the data is randomly partitioned, different results can be obtained due to the characteristics of the expanded dataset. Therefore, to avoid skewing results due to the dataset and focus on the good traits of the automatic feature extraction methodology used in this computer vision methodology and the machine learning model’s real data fitting, several iterations are performed to compute average error metrics to see how good the model can fit the data. The mean weight of the 129 samples in the dataset is approximately 45.1 g

##### Linear Regression (LR)

Manual measurements gave average MAE values ranging from 17.9771 g to 20.6684 g and average R2 values ranging from 0.8305 to 0.8501, while computer vision measurements gave average MAE values ranging from 24.2546 g to 39.1317 and average R2 values ranging from 0.2987 to 0.6872.

Comparing best features, the length–weight relationship gave an average MAE of 17.9771 g and an average R2 of 0.8501 and the mean–weight relationship gave an average MAE of 24.2546 g and an average R2 of 0.6872. Comparing those errors against the mean of the weight distribution for the 129 samples, which is 45.0657 g, the relative errors of the length–weight relationship and mean–weight relationships were 40% and 54%, respectively, which gives a difference of 14% relative-to-mean dataset weight error between them.

##### Power Model (PM)

Notably, the power model outperformed the linear regression, achieving, for the manual measurements, lower MAE (8.01491 g–11.0151 g) and higher R2 (0.8831–0.9187). For the automated features, the model achieved average MAE of 18.1041–33.847 g and average R2 of 0.0606–0.6575.

Comparing best features, the length–weight relationship gave an average MAE of 8.0149 g and an average R2 of 0.9182, and the mean–weight relationship gave an average MAE of 18.1041 g and an average R2 of 0.6575. Comparing those errors against the mean of the weight distribution, the relative errors of the length–weight relationship and the mean–weight relationships are 18% and 40%, respectively, which gives a difference of 22% relative-to-mean dataset weight error between them.

##### Multilayer Perceptron (MLP)

The multilayer perceptron model consistently exhibited superior performance, yielding the lowest average MAE (5.7754 g–6.7985 g) and highest average R2 (0.937–0.9625) for the manually extracted features. It also outperformed the other models using the image processing extracted features, providing average MAE values ranging from 15.6344 to 272.7325 g and average R2 values ranging from 0 to 0.724.

Comparing best features, the length–height–weight relationship gave an average MAE of 5.7754 g and an average R2 of 0.9625, and the mean–weight relationship gave an average MAE of 15.6344 g and an average R2 of 0.724. Comparing those errors against the mean of the weight distribution, the relative errors of the length–height–weight relationship and the mean–weight relationships are 13% and 35%, respectively, which gives a difference of 22% relative-to-mean dataset weight error between them.

#### 4.1.2. Experiments Using the Partitioned Dataset and the Controlled-Light Conditions Samples

The same process used to generate the linear regression, power function, and multilayer perceptron models applied to the manually and automatically extracted features is carried out, but only for the 112 samples that were generated under better lighting conditions. Again, as the data are randomly partitioned, different results can be obtained due to the characteristics of the expanded dataset. Therefore, to avoid skewing results due to the dataset and to focus on the good traits of the automatic feature extraction methodology used in this computer vision methodology and the machine learning model’s real data fitting, several iterations are performed to compute average error metrics to see how good the model can fit the data. The mean weight of the 112 samples in the dataset is approximately 15.2 g.

##### Linear Regression (LR)

Manual measurements gave average MAE values ranging from 2.8058 g to 2.9688 g and average R2 values ranging from 0.5417 to 0.5892, while computer vision measurements gave average MAE values ranging from 3.4169 g to 4.4144 and average R2 values ranging from 0.2218 to 0.4675.

Comparing best features, the height–weight relationship gave an average MAE of 2.8058 g and an average R2 of 0.5417, and the perimeter–weight relationship gave an average MAE of 3.4169 g and an average R2 of 0.4675. Comparing those errors against the mean of the weight distribution for the 112 samples, which is 15.1829 g, the relative errors of the height–weight relationship and perimeter–weight relationships are 18% and 23%, respectively, which gives a difference of 5% relative-to-mean dataset weight error between them.

##### Power Model (PM)

Notably, the power model outperformed the linear regression, achieving for the manual measurements, lower MAE (2.8415 g–3.081 g) and higher R2 (0.5332–0.6303). For the automated features, the model achieved average MAE of 3.3713–4.4583 g and average R2 of 0.1174–0.4946.

Comparing best features, the height–weight relationship gave an average MAE of 2.8415 g and an average R2 of 0.5332, and the perimeter–weight relationship gave an average MAE of 3.3713 g and an average R2 of 0.4494. Comparing those errors against the perimeter of the weight distribution, the relative errors of the height–weight relationship and the perimeter–weight relationships are 19% and 22%, respectively, which gives a difference of 3% relative-to-mean dataset weight error between them.

##### Multilayer Perceptron (MP)

The multilayer perceptron model consistently exhibited superior performance, yielding the lowest average MAE (2.3414 g–2.6392 g) and highest average R2 (0.5649–0.7018) for the manually extracted features. It also gave good results with the image processing extracted features, providing average MAE values ranging from 3.7718 to 195.1027 g and average R2 values ranging from 0 to 0.3478.

Comparing best features, the length–weight relationship gave an average MAE of 2.3414 g and an average R2 of 0.7018, and the perimeter–mean–standard deviation–mode–weight relationship gave an average MAE of 3.7718 g and an average R2 of 0.3478. Comparing those errors against the mean of the weight distribution, the relative errors of the length–height–weight relationship and mean-weight relationships are 15% and 25% respectively, which gives a difference of 10% relative-to-mean dataset weight error between them.

#### 4.1.3. Performance of Feature Extraction Biomass Estimation Methodologies Using the Whole Partitioned Dataset

The last goal of the experiments was to validate the computer vision non-invasive methodology and the characteristics of the dataset. The results that validate the model are those that are closer to manually extracted features, of which the length–weight and height–weight relationships have become a gold standard for allometry relationships in fish.

In the entire dataset, the length–weight relationship gave a 40% error relative to the mean–weight based on the linear regression and 18% based on the power model. The relationship between length and weight in fish adjusts to a power model, thus comparing against this percentage gives us a metric of how good the feature extraction methodology is. The best performance for the computer vision methodologies was achieved by the MLP model, giving a 35% error relative to the mean weight of the dataset. Thus, there is a difference of 17% error relative to the mean of the dataset between the models—or a difference in average MAEs of 7.2 g.

In the case of the dataset of images taken under controlled light conditions, the height–weight power model and its counterpart computer vision power model applied to the perimeter–weight relationship gave an outstanding difference of 3% relative to the mean weight dataset error. Thus, for controlled light conditions, the automatically extracted features method and the manually extracted feature method had an average MAE difference of about 0.5 g. Comparing power models of LxW and PxW, the difference between MAEs is 0.29 g and the relative weight mean is 1.9%. Finally, the best result for the MLP model under the controlled light conditions was an average MAE of 3.7718 g or 24.84% relative to the mean dataset. The difference between MAEs of MLP_P-MEAN-SD-MODE-W_ and LxW is 0.69 g or a relative mean difference of 4.55%.

Figure 10 shows the graphs of the models with the best average MAEs and R2s for the full dataset (controlled light and non-light-controlled conditions). Figure 11 shows the models for the light-controlled dataset only. Table 1 shows the results of all the experiments done with all the dataset models and features.

#### 4.1.4. Experiments with the Two Datasets (Whole Dataset and Controlled Conditions) with the Whole Dataset as the Training Set

The results obtained after applying the same models, features, and datasets but without partitioning the dataset into training and testing (applying the models to all the samples as the training set) and therefore without iterating are shown in Table 2. These experiments were conducted to analyze how well the models and common curves fit the entire dataset for the different lighting conditions and extracted features and not to know the model’s capabilities for generalizing, which was conducted in the first experiment. From those results, the following conclusions can be drawn. Using more data to train the models points toward better results by using feature combinations. Mean and standard deviation give promising results with complex models like MLP, which indicates that these models can improve their biomass estimation by using shape-related features.

### 4.2. Data Sensitivity and Impact of Light Conditions

Subdivision of the dataset into controlled and non-controlled light conditions allowed for a nuanced assessment of the models. Under controlled light, all models, especially the multilayer perceptron, showed improved accuracy, emphasizing the influence of lighting on model robustness. The relative-to-weight dataset mean errors of the best models were obtained to assess if the reduction in error was only due to dataset mean weight variation or also due to light conditions. Given the results, it can be concluded that the models effectively provided more accurate estimations due to controlled light conditions since the relative errors to the mean of the dataset were smaller, showing that better extraction of shape and size is achieved.

### 4.3. Model Generalizability with the MLP Showed Better Performance

The comparison between models trained on the partitioned (train and test sets) subsets revealed the models’ generalizability. Despite the dataset’s limited size, the multilayer perceptron consistently demonstrated robust performance, showcasing its potential for broader applications.

The multilayer perceptron model emerged as a competitive and interpretable solution for non-invasive biomass estimation in tilapia farming and in other species. Its performance surpassed the other applied machine learning models, indicating its potential as a valuable tool for precision aquaculture management. The data tables also show that it can provide better results given more features. Thus, the addition of features like standard deviation or mode could really help the MLP model to give better results, pointing to the use of shape and size for better biomass estimation.

One example of the model fitting a test set searched for high R2 values in test and training sets and low MAE is shown in Figure 12, where red represents the estimated weights. The characteristics of this trained model achieved in the standard deviation–weight relationship are: R2 score training set = 0.7919, mean absolute error = 17.5832, mean squared error = 1104.5208, median absolute error = 4.5979, and explained variance score, R2 = 0.8644. The computer vision model, which was trained to extract features from images, yielded results comparable to the linear regression model applied to the manually extracted features.

One of the best results obtained in the iterations of the MLP model for the controlled light dataset was obtained by using all the features. The result for this iteration was MAE_min_ = 1.7586 g and R2 = 0.7532.

The results for all the models in the partitioned and full datasets are shown in Table 1 and Table 2, respectively.

## 5. Discussion

Section 5.1 contains a discussion of the obtained results and their comparison with the existing literature. We begin by commenting on the morphology–weight relationship, and how the current model provides interpretable features of shape and size that can be used to provide better biomass estimation and an understanding of fish development. Then, in Section 5.2, the experimental results are discussed by looking for better error metrics and comparing them against the state-of-the-art models. In Section 5.3, the results are discussed within the state-of-the-art scope. Finally, practical implications are mentioned in Section 5.4.

### 5.1. Interpretation of Fish Morphology

One intended contribution of this research was to show, precisely, that considering shape factors in addition to size aids in better biomass estimation. As demonstrated by [32], the morphology of fish can undergo different changes as they grow, which can affect the biomass estimation process. The author also recommends considering another dimension (in this case, width) for better estimation since, in some growth stages, using only length and height produces results with lower performance than expected.

With the feature extraction method implemented using a computer vision methodology, many representative statistical measurements of shape and size can be obtained, such as maximum length, perimeter, strip area, mean of perpendicular segments drawn within the fish contour, mode, etc. Furthermore, this research tests a shape and size feature extraction model for biomass estimation. The importance of this lies in having an interpretable model that will allow subsequent studies to explore the relationships between different morphological characteristics of an animal in relation to its weight, unlike deep learning models, which function as highly effective black boxes in prediction but may complicate the extraction of interpretable data that aids in better understanding of morphological and allometric development in fish. It could also help improve the estimation of fish biomass by finding a common morphology of the fish at different life stages, which could be a parameter to consider when using monocular vision since the shape could be indicative of age (and development stage), which was used by [28] for biomass estimation.

As fish and crustaceans grow, they undergo morphological changes like those observed in human body proportions during development. This indicates that relying on a single factor such as length (L) or height (H) may not provide the same validity across different stages of their development, as demonstrated by [32]. Therefore, a computer vision model for analyzing shape and size and their relationship with biomass could be a good alternative for investigating morphological changes and their correlation with biomass at various fish and crustacean growth stages. This approach could help enhance understanding of animal development across different growth stages and aid in detecting diseases, malformations, or malnutrition without the stress or risks associated with manual and invasive handling. This benefit is highlighted in [14], where the issues of disease and malformation detection using computer vision are addressed, and in [16], where non-intrusive methods for biomass estimation in aquaculture are examined.

### 5.2. Model Performance and Standardization

After an exhaustive review of the literature, it can be observed that comparing machine learning and deep learning models in aquaculture environments using only metrics like MAE or RMSE does not truly allow us to appreciate which model can provide better accuracy or results for a specific problem or if it presents good generalization capabilities to other conditions and specimens. An example of the aforementioned statement is provided by [25,29], who obtained MAE metrics of 2.27 g and 3.47 g; however, dividing by the mean weight of the dataset gives MAE/W_mean_ of 3.14 and 2.79%, respectively. All the models are developed for very particular conditions of lighting, species, specimen development stage, quantity of specimens, diets, and the environmental conditions under which they are being raised, as explained in the related works section and stated in reviews [14,15,16,18,38].

The computer vision problem presents significant challenges, as mentioned by [15,18,38], and an open challenge for standardization, as even several datasets that used the same species and similar weights could have used images that were taken with cameras with different resolutions, from different angles, and at different locations, etc. Therefore, it can happen that model A with better MAE, RMSE, and R2 metrics gives worse results than model B in a different environment or even small variations where the metrics were calculated. For this reason, in the following discussion, we will first analyze the results of the experiments obtained in the present research. Given these results and what was observed in the literature review conducted, it can be considered a fundamental and good practice when developing computer vision-based methodologies for estimating biomass to make a comparison between image feature extraction models of fish or crustaceans against machine learning methods that use manually extracted features, as done by [4,31,32]. From the literature review and according to [11], using manually extracted features such as length and height could be the gold standard against which to compare automatic image feature extraction methods and deep learning methods that directly determine the weight of animals from the corresponding images. That is to say, the common machine learning models applied to manually extracted features of the specimen compared against the values obtained using the computer vision methodologies are a good indication of how good the model is since simple machine learning models for relating simple size–weight parameters usually have good performance (R2) under controlled light conditions, as can be seen in Table 1 in [16] and which was actually performed in [29].

To standardize further and eliminate differences between model results when applied to datasets with fish of different sizes, the ratio between the mean absolute error and the average weight of the dataset (MAE/W_mean_) could be used. As observed in [26,27,32], for different weight ranges and as seen in the dataset used here, the MAEs increase with the average weight of the dataset. This makes sense because for two fish of different sizes in the same environment, the bigger fish will have an image with more pixels and after the same segmentation process (e.g., a threshold method), the same proportion error in both images would produce a bigger MAE for the biggest fish. Therefore, using metrics relative to the size of the dataset could provide better results that could be better compared against other methods. More work needs to be done on this topic.

In the comparison made, it was demonstrated that differences in the average mean absolute error of models with manually obtained characteristics versus those obtained by the image processing algorithm could be 0.5 g for a power model and 0.69 g for the more feature- and environment-consistent MLP model or in relation to the mean dataset weight, 3% and 4.55% ((MAE_manual_ − MAE_CLF_)/W_datasetMean_). Therefore, these results strongly suggest the validity of the feature extraction method developed in [34] and implemented in a computer vision pipeline for biomass estimation in this work. Investigations could be conducted throughout the fattening or development process in aquaculture ponds or research centers, obtaining histograms that represent the shape and size of different specimens (fish and crustaceans) and thus making comparisons and gaining new insights into morphological and allometric development of animals from an approach of representative statistical parameters of shape and size obtained through a signature function.

### 5.3. Comparative Analysis

To compare this computer vision-based methodology, whose core is an interpretable method for extracting statistical features of shape and size, against the state of the art, various methodologies that have been developed for fish and shrimp biomass estimation are presented in Table 3, along with the metrics provided in the corresponding articles. It should be noted that different authors provide different metrics, datasets, and different architectures for image capture. Here, the research is divided into two main categories. The first is those in which test images were taken by cameras out of the water, and the second is in those studies where images or videos were taken with underwater cameras. In this table, the present method’s metrics shown are those of the computer vision methodology comprising the chord length function feature extraction methods plus the multilayer perceptron model for biomass estimation.

Table 3 shows that the range of values for the methods in which the dataset was developed with a camera out of the water for the MAE/W_mean_ ratio is 2.79–7.09%. For models based on underwater cameras, it was from 0.9 to 12.57%, resulting in a total range of 0.9–12.57% for all models. This corresponds to a range of reported MAEs of 0.04 g–60.16 g. Considering one of the best results obtained in this study, the models tested based on CLF could reach up to 11.6% (for some of the multilayer perceptron iterations) corresponding to an MAE of 1.76 g; placing the model within the range for underwater camera systems, but below those with cameras out of the water (in this case the fish could be inside the water or outside in a prepared setup). However, the average results (from 50,000 and 300 iterations) in the test set and the curve fitting results for the entire dataset as a training set reach a minimum for the MLP_Perimeter-Mean-Standard Deviation-Mode_ characteristics of MAE_5000_/W_mean_ = 3.39 g/15.18 g = 20.33%. Now, this does not necessarily imply that these models are more accurate than the one in the present work, given that in Poonpat Poonnoy (2022), where an MAE/W_mean_ of 3.14% is presented for their CNN-based model corresponding to an MAE of 2.27 g, a comparison is also made for the same dataset with a method based on area–weight ML relationships with least squares, resulting in an MAE/W_mean_ of 3.7% corresponding to an MAE of 2.69 g, which does not show a difference of more than 1% corresponding to a difference of 0.44 g between the MAEs of the methodologies. By performing the same process with our models for controlled lighting conditions like those in the mentioned article, the following results are obtained for comparison.

The average result of our area-based model (the area is not the total pixel area of the fish but of the strips drawn within the contour, which does have a relationship with the true area) gives an MAE/W_mean_(%) = 3.39 g/15.18 g = 22.33% for our dataset (this is the average MAE value for the power model after 50,000 iterations), and given the value provided by the MLP and a combination of the four factors in the test set of MAE/W_mean_(%) = 3.77 g/15.18 g = 24.83%, which indicates that the difference between models of 2.5% corresponding to a difference of MAEs of 0.36 g between methodologies. For both cases, a difference of percentages no greater than 3% is observed. Given the consistency of the MLP model in outperforming the others, it is possible to give similar or even better results. Now, given the literature review conducted, the results of which are presented in Table 3 with the metrics and datasets used, it is concluded that for those models based on image feature extraction, a fair comparison is to compare them against machine learning models based on manual feature extraction in the same dataset. As already mentioned in the present research, with the multilayer perceptron model for 300 iterations of the test set and the entire dataset, a difference of percentage less than 4.55% can be obtained.

### 5.4. Practical Significance

In a broader context, the proposed methodology exhibited noteworthy advantages over existing computer vision-based biomass estimation approaches. The interpretability of the signature function-based features provides insights into morphological and allometric relationships that could enhance understanding and generalizability across aquaculture species. With the results obtained, the method can already be used in controlled environments for non-invasive sampling in low-height containers, both for a single specimen or for several specimens at low density and shallow depth. This is already a practical outcome since the stress generated by taking an image in a container with water is much lower than caused by removing the fish from the water and placing it on a scale. These results pave the way for future research in refining the proposed methodology, exploring its application to diverse aquaculture species, and investigating the impact of environmental factors on biomass estimation for industrial and academic in situ setups.

## 6. Conclusions

This work advances the development of a non-invasive fish biometrics methodology based on the feature extraction algorithm developed by [34] validating its biomass estimation capabilities on fish in conjunction with a machine learning model and a top-placed camera. Three ML models with different complexities were tested and the multilayer perceptron emerged as the most consistent under different features and environmental conditions. The resulting metrics showed that the resulting computer vision methodology gives very good results against the manually extracted features and is at par with other alternatives. Given its interpretable nature, the quantity of shape and size features that can be extracted can be applied to different species. Thus, it is a good choice that can be used in a computer vision pipeline for obtaining real-time non-invasive biometrics for aquaculture and biological studies. In such a manner, a better comprehension of morphology-biomass relationships can be studied for the different life stages of several cultured species.

This work had the following limitations that could affect the provided results: a low-resolution camera was employed and the dataset comprised 129 specimens; however, the average total weight was about 45 g, which is not close to the biggest size of tilapia reported in the literature. Simple preprocessing steps were used, and threshold segmentation techniques were chosen by the authors based on visual inspection of the resulting segmentations. Only seven threshold techniques provided by a Python library were tested, with the triangle threshold binarization giving the best results. Thus, the binarization in this work was done by manual selection of one of seven options for each fish.

Future research needs to be conducted to validate biomass estimation using other species. Additionally, better preprocessing and segmentation techniques would be needed to apply the feature extraction plus machine learning methodology steps in real setups in a fully automated computer vision pipeline. This would contribute to the development of a powerful tool with good potential for aquaculture-cultivated species proposing new challenges.

## Figures and Tables

**Figure 1 animals-14-01850-f001:**
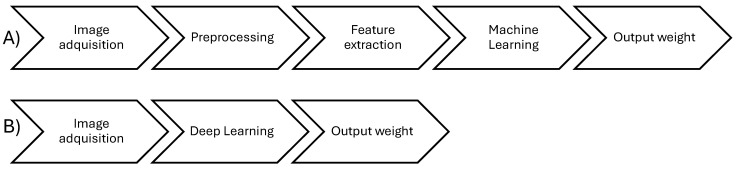
General stages of a computer vision weight estimation methodology. (**A**) computer vision methodology based on image processing feature extraction and machine learning regression models. (**B**) Weight estimation based on deep learning computer vision methodology.

**Figure 2 animals-14-01850-f002:**
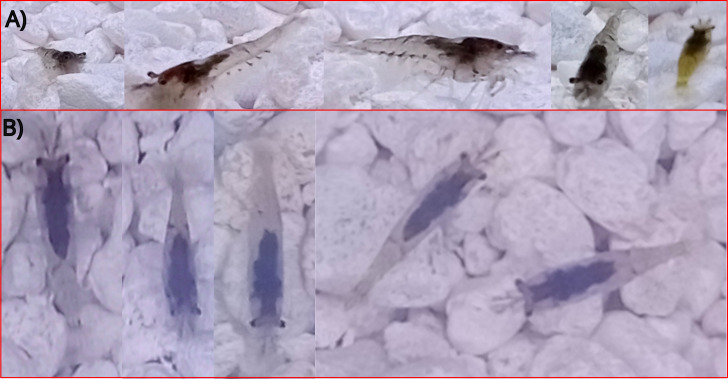
Neocardinas (freshwater shrimps) as seen from (**A**) low angles with respect to the plane parallel to the bottom and (**B**) high angles in relation to the plane parallel to the bottom.

**Figure 3 animals-14-01850-f003:**
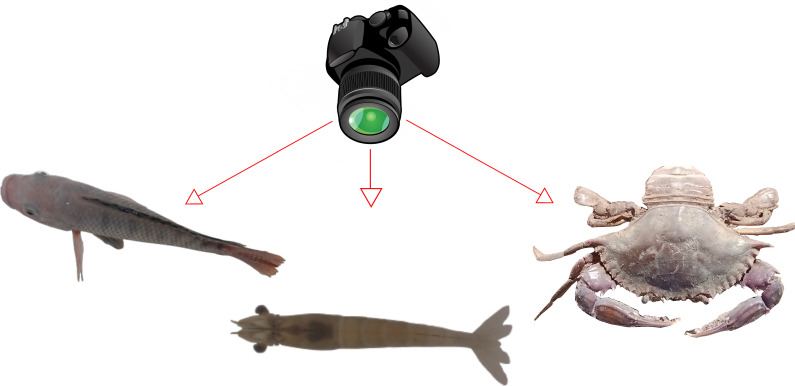
Three common cultivated species as seen from high angles with respect to the plane parallel to the bottom Surface.

**Figure 4 animals-14-01850-f004:**
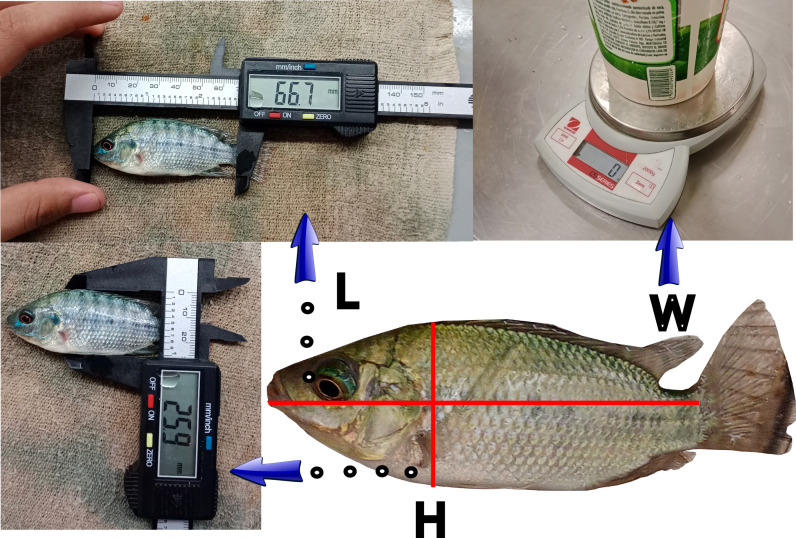
Traditional methods for sizing and weighing Nile tilapia (*Oreochromis niloticus*).

**Figure 5 animals-14-01850-f005:**
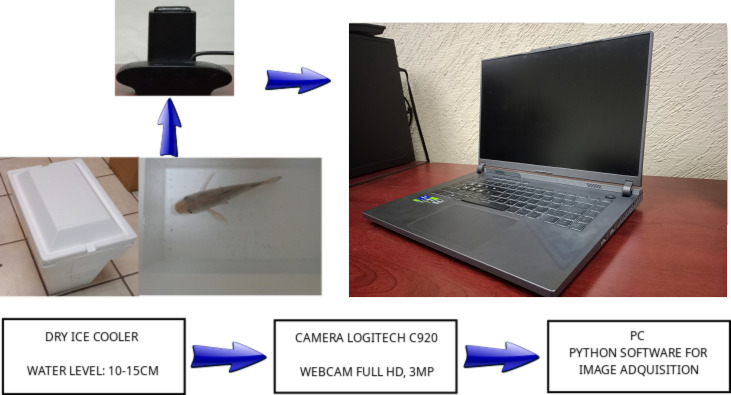
Image acquisition setup.

**Figure 6 animals-14-01850-f006:**
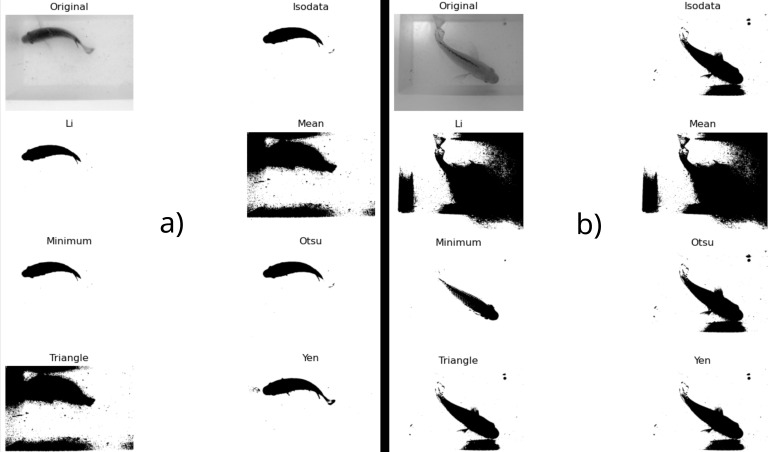
Segmentation threshold methods applied to a sample fish (**a**) from the controlled-light dataset, and (**b**) from the non-light-controlled dataset.

**Figure 7 animals-14-01850-f007:**
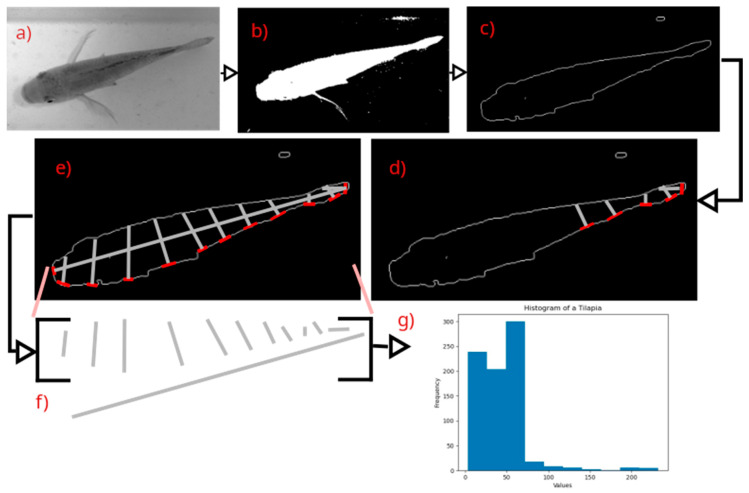
Stages of the shape analysis method using a chord length function. Subfigure (**a**) shows the grayscale image, (**b**) shows the segmented image, (**c**) shows the extracted contour after morphological operations, octal thinning algorithm and Canny edge detection algorithm, (**d**,**e**) shows the drawing of the perpendicular line segments contained in the fish contour, (**f**) shows the extracted perpendicular line segments stored in a container or array, and (**g**) shows the representative histogram of the fish’s contour.

**Figure 8 animals-14-01850-f008:**
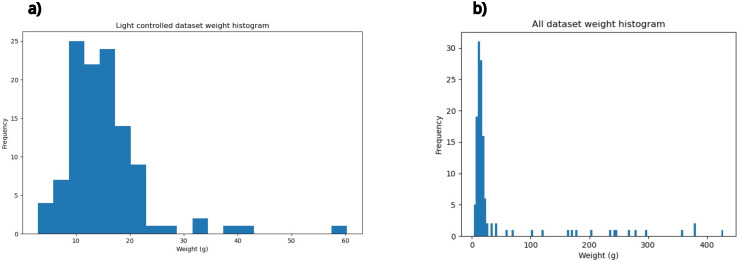
Histograms of weights of the light-controlled dataset (112 samples) shown in subfigure (**a**) and the whole dataset (129 samples) shown in subfigure (**b**).

**Figure 9 animals-14-01850-f009:**
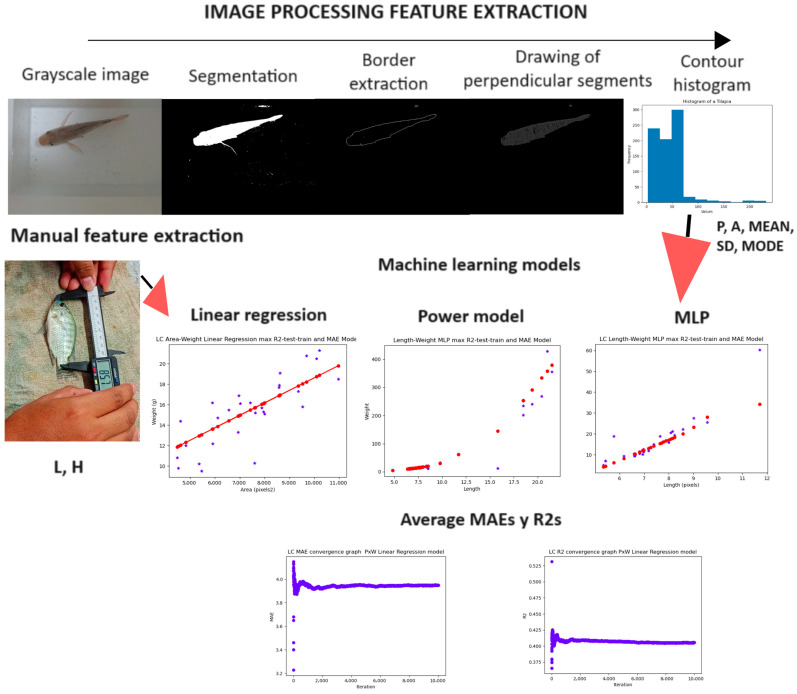
The manually extracted features are compared to the extracted image-processed features by feeding machine learning models and obtaining the MAE and R2 for the full dataset or the average MAE and average R2 for the partitioned dataset. The red line and dots on the three machine learning model estimation graphs represent the predicted values by the models, while the blue stars indicate the real values.

**Figure 10 animals-14-01850-f010:**
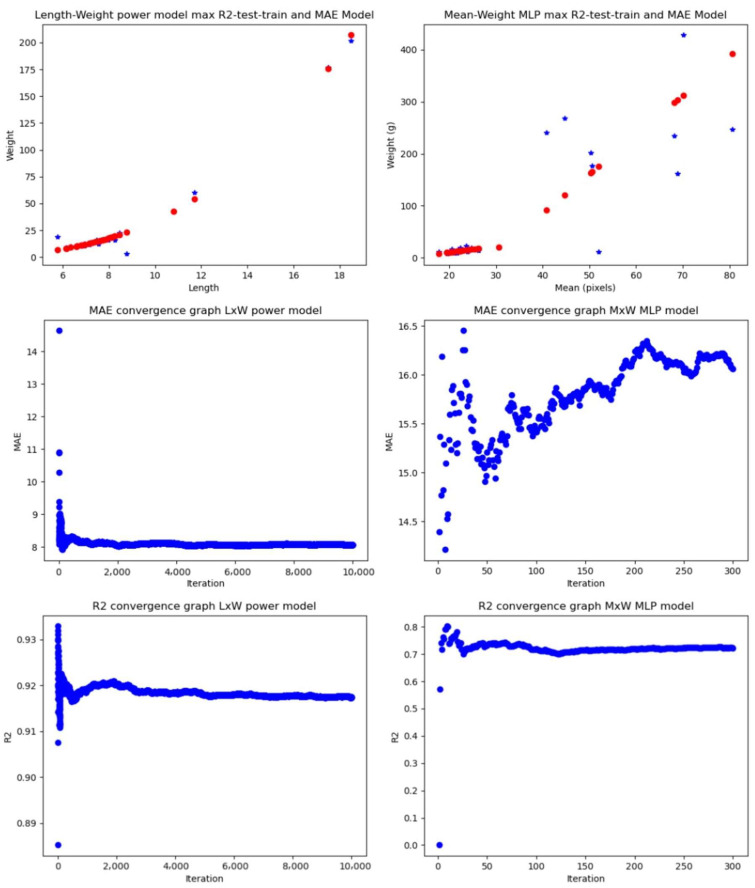
Best-performing models based on the whole dataset. Manual extracted features + machine learning (**left**) and image processing + machine learning (**right**). The blue stars are the real values measured by balance and the red dots are the predicted values by the model.

**Figure 11 animals-14-01850-f011:**
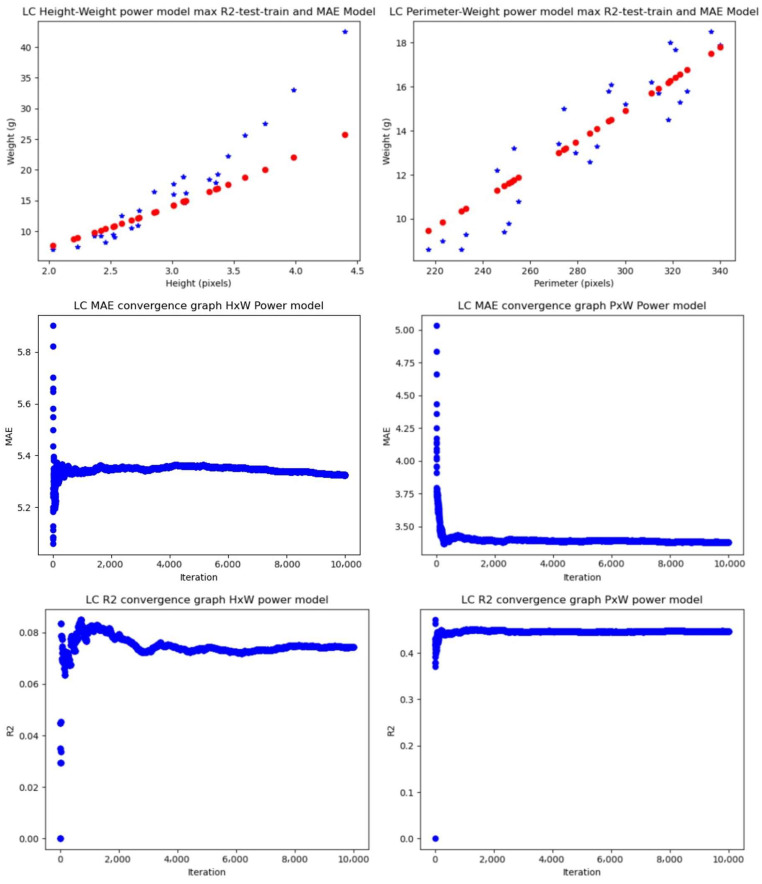
Best-performing models based on the controlled-light dataset. Manual extracted features + machine learning (**left**) and image processing + machine learning (**right**). The blue stars are the real values measured by balance and the red dots are the predicted values by the model.

**Figure 12 animals-14-01850-f012:**
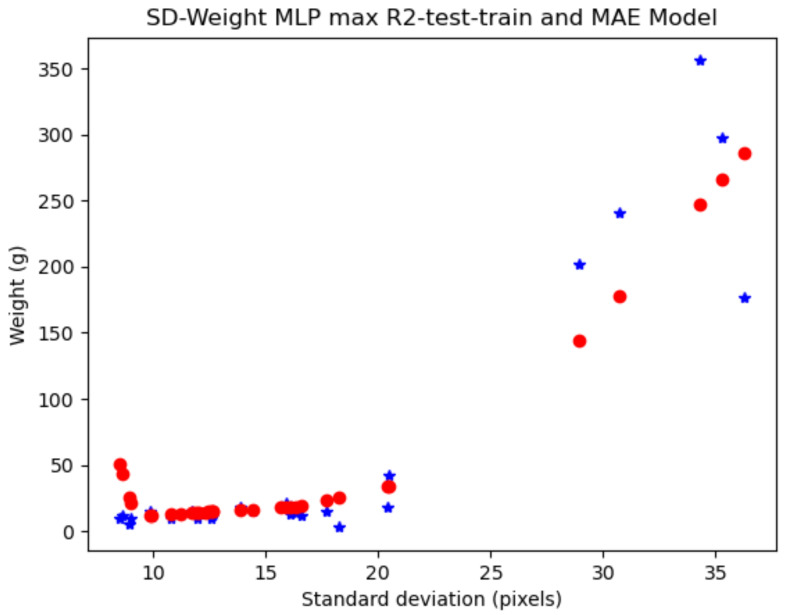
Example of good model results for all the data (129 samples), partitioned into training and test datasets, based on standard deviation. The blue stars are the real weight values and the red dots are the predicted weight values by the model.

**Table 1 animals-14-01850-t001:** Test dataset. Averages of MAE in grams (g) and R2 for all models, image sets, and features tested. The chosen characteristics were length (L), height (H), perimeter (P), area (A), Mean, Mode, and standard deviation (SD) associated with weight (W).

Features–Weight Relationship	Average MAE Linear Regression 50,000 Iterations	Average R2 of Linear Regression	Average MAE Power Model 50,000 Iterations	Average R2 of Power Model 50,000 Iterations	Average MAE of MLP 300 Iterations	Average R2 of MLP 300 Iterations
Dataset with images taken under non-light-controlled conditions						
Machine learning models applied to manually extracted features						
L-W	17.9771	0.8501	8.0149	0.9182	6.7985	0.937
H-W	20.6684	0.8305	11.0151	0.8831	6.748	0.9413
LH-W	18.451	0.8467	8.228	0.9187	5.7754	0.9625
Machine learning models applied to the features extracted by the image processing algorithm based on the CLF function						
P-W	39.1317	0.2987	27.8686	0.2419	43.0031	0.2859
A-W	26.8086	0.5578	21.6221	0.4992	272.7325	0
MEAN-W	24.2546	0.6872	18.1041	0.6575	15.6344	0.724
SD-W	31.4924	0.6201	23.1221	0.5278	17.9506	0.695
MODE-W	37.8471	0.4439	33.847	0.0606	23.9877	0.5893
P-MEAN-SD-MODE-W	25.3608	0.6526	20.9863	0.6137	39.2222	0.3325
Dataset with images taken only under controlled light conditions						
Machine learning models applied to manually extracted features						
L-W	2.9688	0.5892	3.081	0.6303	2.3414	0.7018
H-W	2.8058	0.5417	2.8415	0.5332	2.6392	0.5649
LH-W	2.9229	0.5567	3.0298	0.5849	2.4794	0.5696
Machine learning models applied to the features extracted by the image processing algorithm based on the CLF function						
P-W	3.4169	0.4675	3.3713	0.4494	4.0664	0.2105
A-W	3.9372	0.4051	3.3865	0.4946	195.1027	0
MEAN-W	3.9289	0.4058	3.9086	0.3798	4.0473	0.3113
SD-W	4.0587	0.3154	4.0089	0.28753	4.0833	0.3094
MODE-W	4.4144	0.2218	4.4583	0.1174	4.3308	0.285
P-MEAN-SD-MODE-W	3.6538	0.4027	3.5439	0.42498	3.7718	0.3478

**Table 2 animals-14-01850-t002:** Complete dataset. MAE in grams (g) and R2 for all models, image sets, and features tested. The chosen characteristics were length (L), height (H), perimeter (P), area (A), Mean, Mode, and standard deviation (SD) associated with weight (W).

Features–Weight Relationship	MAE Linear Regression (129 Fish)	R2 of Linear Regression (129 Fish)	MAE Power Model (129 Fish)	R2 of Power Model (129 Fish)	MAE of MLP (129 Fish)	R2 of MLP (129 Fish)
Dataset with images taken under non-light-controlled conditions						
Machine learning models applied to manually extracted features						
L-W	17.3654	0.8859	7.3354	0.9321	5.6037	0.9679
H-W	20.0136	0.8719	10.5337	0.8961	5.8034	0.9722
LH-W	17.5856	0.8868	7.5223	0.9345	4.3594	0.9817
Machine learning models applied to the features extracted by the image processing algorithm based on the CLF function						
P-W	38.0444	0.3122	27.0306	0.2148	39.3803	0.3116
A-W	25.5733	0.6085	20.3838	0.5259	274.8166	−22.4163
MEAN-W	23.381	0.7254	17.4008	0.6987	12.6084	0.8557
SD-W	30.5311	0.6622	22.5198	0.5423	15.1603	0.8364
MODE-W	36.4037	0.4225	33.5415	0.0217	20.0281	0.7561
P-MEAN-SD-MODE-W	22.4546	0.7476	17.116	0.694	15.2622	0.8597
Dataset with images taken only under controlled light condition						
machine learning models applied to manually extracted features						
L-W	2.6067	0.4734	2.7272	0.295	1.5593	0.8461
H-W	2.5408	0.4815	2.6513	0.4215	2.4287	0.5255
LH-W	2.4464	0.4929	2.5813	0.3821	1.3901	0.8862
Machine learning models applied to the features extracted by the image processing algorithm based on the CLF function						
P-W	3.1492	0.418	3.1765	0.379	3.9628	0.2174
A-W	3.5161	0.3159	3.1286	0.3423	195.184	−803.8525
MEAN-W	3.6511	0.3414	3.6799	0.29	4.0098	0.3026
SD-W	3.907	0.3119	3.8882	0.2919	3.9229	0.3201
MODE-W	4.1732	0.1742	4.3413	0.0846	4.3099	0.1633
P-MEAN-SD-MODE-W	3.0681	0.4392	3.0913	0.3997	3.3923	0.4221

**Table 3 animals-14-01850-t003:** Comparison between the different biomass estimation methods. Many metrics in the last column were not given in the corresponding articles and were calculated based on the data they provided.

Work	# of Specimens	# of Samples (e.g., Photos)	Average Weight of Datasets (AWD)—g	Best Presented Related Metrics	ErrorMetric/ Wmean (%)
Pictures and/or videos taken out of the water					
[21]	122	NA	19.06	<4%	<4%
[22]	75	77	184.82–207.35 g	3.25%	3.25%
[23]	NA	1072	Range of datasets: 200–1000 g, 1000−2000 g	R = 0.9828, MARE = 5.58%	MARE = 5.58%
[24]	1400	1400	150–1000 g	MAPE = 4.28%	MAPE = 4.28%
[25]	10	10	124.51	MAE: 3.47 MEDAE: 2.45 MSE: 19.681 RMSE: 4.4363 MAPE: 0.0296 Mean Percentage Error = 2.82%	MAPE = 0.0296 Mean Percentage Error = 2.82% MAE/W_mean_(%) = 2.79%
[26]	25	600	W_mean600_ = 115.88	RMSE = 9.19 ± 3.74 g, MAE = 6.06 ± 3.64 g, MARE = 5.18 ± 3.08%, MXAE = 8.87 ± 3.26 g, MXRE = 0.12 ± 0.12%, N15R2 = 0.77 ± 0.10, N60R2 = 0.96	MAE/W_mean600_ = 5.6%
[27]	Test = 170	1700	423.81 g	Accuracy = 93.01 ± 3.11%	MAE/W_mean_ = 7.09%
[29]	3832 Test = 958	3832 Test = 958	72.2 g	R2 = 0.99 MAE = 2.27 g RMSE = 3.59 g MRE(%) = 0.05 MAPE(%) = 4.09 g Power área model for the same dataset MAE of 2.69 g giving a MAE/W_mean_ = 3.73%.	MAE/W_mean_ = 3.14%
CLF_CV_MODEL BASED ON MLP (OUR PROPOSAL)	129	Approx = 3870 129 selected	All_dataset = 45.1 g CL_dataset = 15.2 g	Best average MLP CL data PERIMETER-MEAN-SD-MODE: R2_score training set_ = 0.44 avg_MAE_300_: 3.67 g avg_MEDAE_300_: 2.22 avg_MSE_300_: 44.27 avg_RMSE_300_: 6.30 avg_MAPE_300_: 0.31 avg_R2_300_: 0.35 Best achieved score in one iteration: MAE_min_ = 1.7586 g and R2 = 0.7532	Best average MLP_CL MAE/W_mean_: 24.8% MAE/W_mean-DifMACV_ = 4.55%. MAE/W_mean-DifMACV_ = 0.69 g Best achieved score in one iteration: MLP_CL MAE/W_mean_: 11.58%
Pictures and/or videos taken underwater					
[21]	NA	NA	NA	<5%	<5%
[4]	150	NA	NA	MAE = 24.5 g R2 = 0.9736	MAPE = 11.24% MAPE-DifMACV = 1.63% MAE-DifMACV = −13.1 g
[28]	90	Total_images = 4287 Split Train = 60% Test = 40%	Max = 482.24 g Wapprox = (max + min)/2 = 324.35 g	MAE of 40.78 g, R2 of 0.74. average weight error of only 30.30 (±23.09) grams	MAE/MaxW =8.5% MAE/Wapprox = 12.57%
[30]	Training = 20 Test = 6	Total = 6400 Used	W_mean_ = 0.96 g	R2 = 0.96 RMSE = 0.05 g MAE = 0.04 g	MAE/W_mean_ = 4.17%
[31]	Sampled = 190	AI estimated = 124	606 g	W_meanDiff_ = 3.6%	W_meanDiff_ = 3.6%
[32]	550	8 videos	455.38 g (1.62 kg/m^3^) and 460.11 g (6.50 kg/m^3^).	RMSE =34.90 g MRE = 7.41% in H-W. Multilayer Perceptron MAE = 60.16 g, RMSE = 70.71 g R2 = 0.	(Wmean-Wpred)/Wmean1.62 kg/m^3^ = 4.15% (Wmean-Wpred)/Wmean6.50 kg/m^3^ = 0.9% RMSE/Wmean1.62 kg/m^3^ = 7.7%

## Data Availability

Images and data used for this work are available at Fernando Joaquín Ramírez-Coronel, Edgard Esquer-Miranda, Oscar Mario Rodríguez-Elías, 16 May 2023, “Nile Tilapia (Oreochromis Niloticus) dataset for non-invasive fish biometrics with computer vision models”, IEEE Dataport, doi: https://dx.doi.org/10.21227/kq7f-4h27.

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
