# Peer review of "Non-Invasive Fish Biometrics for Enhancing Precision and Understanding of Aquaculture Farming through Statistical Morphology Analysis and Machine Learning"

_animals, 2024, doi:10.3390/ani14131850_

Round 1
Reviewer 1 Report
Comments and Suggestions for Authors
The manuscript should be significantly shorter.
The references need a lot reformating
lines 90-92 need a reference
line 118 you mean section 6 and not 5
lines 164-165 Oreochromis mykis does not exist to my knowledge
section2 needs a reference per sentence (lines 121-146)
Lines 310-322 can just be taken out
section 4.1 no need for an explanatory text
lines 687-688 do not understand the sentence please explain
section 5.1 lines 714-724 can just be taken out
lines 745-759 English needs improving
line 783 "The authors also consider"... This is a discussion not a matter of opinions, please re phrase accordingly
section 5.4 should be part of the conclusion and section 6 should be shorter
Issues with the methods
why did you only use fish at a depth of 10 to 15 cm, this is not a realistic situation?
why did you not do a more robust validation with completely new fish that were not part of the 129 samples
why did you not do a biometry by hand to verify all the data?
How did you select the Features (under what criteria)?
How does the algorythm perform when the fish have deformities?
section 4.2 Why did you not finetune the system? it reads like you are just insisting but not attempting to improve.
The bigger the fish the less robust is the method, so how do you propose to improve from here? or will you set a size limitation?
Comments on the Quality of English Language
There some comments already in the previous section
Author Response
The responses are in the attached file.
Please see the attachment
Thank you.

Reviewer 2 Report
Comments and Suggestions for Authors
The application of computer vision techniques in agriculture is a growing trend. In aquaculture, numerous studies are being conducted with the aim of producing scientifically relevant information to contribute to the development of technologies that make the activity more viable, improve the quality of life for the workforce, and promote animal welfare. I consider the work relevant and innovative, although the dataset used is small.
The authors employ an innovative and highly effective approach by utilizing feature extraction techniques based on signature functions. The careful selection of characteristics to be analyzed and the meticulous design of the signature function demonstrate a profound understanding of the data complexity and analysis objectives. Furthermore, the application of these techniques results in precise and reliable estimations, highlighting the potential of this methodology to address practical problems, such as estimating fish body weight. This work not only significantly contributes to advancing the field but also paves the way for future applications and research in other areas that could benefit from this innovative approach.
It is not clear where the morphometric measurements were manually collected; it would be relevant to include a photo showing each measurement.
Regarding the database, there appear to be some outliers; was any methodology applied to verify if these data are influential? Also, is there a need to include these data points in the analysis?
With respect to the small dataset, how do the authors validate the work? Would a larger dataset be necessary?
The paper lacks an explanation of the applicability in fish farms, considering that the collection environment has clear water, which is not common in most fish farms.
It would be relevant to perform correlation analysis between the collected variables with the observed and predicted body weight, as well as those extracted through segmentation.
Comments on the Quality of English LanguageI suggest that the paper undergo English language correction before being published."
Author Response

(The authors gave the same response as above.)
